# Optical Emission Detector Based on Plasma Discharge Generation at the Tip of a Multimaterial Fiber

**DOI:** 10.3390/s20082353

**Published:** 2020-04-21

**Authors:** Clément Strutynski, Lionel Teulé-Gay, Sylvain Danto, Thierry Cardinal

**Affiliations:** Institute of Chemistry of the Condensed Matter of Bordeaux (ICMCB), University of Bordeaux, 33608 Pessac, France; lionel.teule-gay@u-bordeaux.fr (L.T.-G.); sylvain.danto@u-bordeaux.fr (S.D.); Thierry.Cardinal@icmcb.cnrs.fr (T.C.)

**Keywords:** fiber design and fabrication, multi-material fiber, hybrid fibers, gas sensor, plasma

## Abstract

Experimental development of a compact optical emission detector based on the assembly of a polymer-metal and a standard silica fiber is presented in this paper. This device is exploited in a proof-of-principle experiment for gas detection application by means of plasma spectroscopy in the visible-Near Infrared spectral region. A multimode fiber (MMF) is associated with a functional hollow dual-electrodes elongated structure fabricated by the direct preform-to-fiber homothetic co-drawing. A potential of 1.5 kV is applied between the two electrodes embedded inside the composite cladding, which generates an atmospheric pressure dc glow discharge at the tip of the fiber bundle. The emitted light is then collected by the MMF for optical diagnostics. Probing of different atmospheres is presented at the end of this study.

## 1. Introduction

There is currently a tremendous need in the development of hybrid, multi-purpose optical fibers featuring simultaneously multiple functionalities. The fabrication of elongated structures combining special optical, electrical, or mechanical assets allows for the production of fully integrated and more compact useful tools for signal-processing or sensing applications [1]. Among the opportunities with strong scientific interest accessible with this class of composite builds, is the merging of fiber technology and plasmas. Previous studies have already reported on the development of fibered-devices using plasmas for medical applications (dentistry, endoscopy, oncology, etc.) [2,3,4], and more generally for optical diagnostics in remote, confined, or harsh environments [5,6,7]. Usually they make use of a small, optical, fiber-based apparatus at the end of which a plasma is carried or generated. For instance, all-fibered Light Induced Breakdown Spectroscopy (LIBS) systems have been developed for various all-optical distant compositional analysis [8,9]. A high energy ultrashort pulse is guided through an optical fiber and focused on a sample (solid, gas, or liquid), which generates a plasma which emission spectrum can be analyzed to identify the atomic or molecular species composing the probed material. A major drawback in this configuration is the need for a focusing stage at the extremity of the equipment. Additionally, several limiting factors must be taken into account concerning the fiber delivering the high-power laser signal including its numerical aperture, damage threshold, and intermodal dispersion. Another class of instruments combing elongated structures and plasmas are micro-plasma jets [2,3,4]. They make use of flexible hollow-core fibers in which a controlled gas flow circulates. A potential is applied between two electrodes (internal and external) placed close to the extremity of the elongated tube, which generates a plasma stream. Alternatively, a single coiled-electrode can be used with a radio frequency (RF) generator for induction-based plasma generation. Tailoring of the capillary dimensions allows for fine control of the size of the glowing excited gas flow [4] to better suit the targeted utilization (tissue healing, apoptosis, bacteria elimination, etc.). This kind of apparatus is, however, almost exclusively used for treatments but not for diagnostics.

Despite the strong interest of merging plasmas with fibers for sensing applications, no multi-material fiber-based systems have yet been considered for this topic. Such developments rely on the direct preform-to-fiber co-drawing of intricate composite structures combining ultraviolet (UV)-visible-near infrared (NIR) transmitting glasses, metals, and polymers [10,11]. A significant challenge, however, resides in the matching of the softening temperatures of the different materials involved in the macroscopic preform design for viscosity management during thermal drawing. An alternative route to circumvent this issue and form compact hybrid devices is to put together several fibers with a specific singular purpose [12]. In this study, taking full advantage of both the technological maturity of silica fibers (low optical losses, advanced processability, etc.) and the excellent shaping ability of polymers (easier machining, good mechanical properties in the fiber form, etc.), a hollow polymer-metal fiber and a multimode fiber (MMF) are associated with the goal to produce an optical emission detector based on plasma spectroscopy. The potential of this proof-of-principle device for continuous monitoring of the atmosphere is then assessed through gas detection experiments.

## 2. Results and Discussion

Due to their greatly advanced technological maturity, silica fibers offer an undisputed platform for light manipulation (management of propagation, dispersion, polarization, etc.) and system integration (Bragg gratings inscription, splicing, etc.). In the meantime, polymers also possess interesting assets for fiber development. Intricate multi-material preform profiles can be easily built from those materials using standard machining methods. Additionally, the fabricated fibers are easier to process and manipulate (connection of embedded electrodes, bending, etc.) due to their mechanical properties. In this case, we explore the potential of combining those two classes of fibers for gas detection applications. The experimental configuration relies on the association of a silica MMF with a hollow, elongated dual-electrodes composite fiber fabricated by the direct preform-to-fiber drawing method (see Figure 1a).

A central hole of 5 mm surrounded by two 1.5-mm holes are mechanically drilled in a 19-mm diameter and 75-mm long polyether-sulfone (PES) rod. Tin electrodes are inserted inside the lateral holes, while the one at the center remains empty. The thermal co-drawing ability of the materials is assessed using a dedicated three-meters-high optical fiber draw tower. The preform is placed in the furnace and the temperature is gradually ramped up to ~300 °C to initiate the drawing process. Then the rod is slowly fed into the furnace while the drawing parameters are continuously monitored to produce a fiber with a targeted central hole with a diameter of ~150 µm. Tens-of-meters of composite hollow dual-electrode fibers are fabricated in that manner. PES is chosen in this case as the cladding material since it is a fiberizable polymer exhibiting one of the highest glass transition temperature (*T_g_* ~225°C). This provides a moderate resistance to the potential heat effects for the fabricated fiber that can occur during plasma discharges where high (~kV) tensions are involved. Moreover, transparency of PES in the visible area provides practical advantages for sample processing (electrode connection, etc.) or manipulation. Tin is selected for the electrode material because its fusion (*T_m_* = 232 °C) matches well with the thermal stretching temperature range of PES. Association of those materials for fiber drawing allows for good viscosity management and precise conservation of the preform profile during the homothetic process, as demonstrated in the literature [10]. Tin also exhibits a good conductivity (around 8 × 10^6^ S m^−1^ [13]) while PES possesses a rather high dielectric strength (approximately 15.5 kV mm^−1^ [14]), which is necessary to sustain a high voltage discharge. Description of the composite structure fabricated for this study is shown in Figure 1a. It is composed of a ~620 µm cladding with a central hole of ~150 µm surrounded by two ~50 µm tin electrodes themselves separated by a ~300 µm gap. Dimension of the inner hole is tailored so that most of the commercially available fibers (Silica single mode fibers, multimode fibers, graded-index fibers, and Fluoride IR patch cables, etc.) possessing standard 125-µm external diameter can fit in it, which allows this composite structure to be used as a functional cladding with a wide variety of common fibers. Then, ~5 cm long portions of this PES/Sn fiber are selected for the assembly of the optical emission detection device. A silica MMF (FG050LGA, Thorlabs) with core/cladding dimensions of 50/125 µm is stripped from its acrylate coating and inserted inside the central hole (see Figure 1b). Numerical Aperture (NA) of the collection fiber is 0.22. Particular care is taken to precisely align the extremities of the two fibers, which are, subsequently, epoxy-glued together. Cross-sectional observation of the tip of the assembly with an optical microscope in transmission mode is shown in Figure 1b. Extremities of both the PES/Sn and silica fibers are found to be on the same plane, which confirms the good alignment of the structure. An external adjacent connection of the embedded electrodes is then performed. The PES cladding is stripped away (see Figure 1c) with a scalpel blade on a ~ 500 × 500-µm surface and 25-µm copper wires are attached to the exposed Sn electrodes with silver paint. In this instance, the use of the polymer for the composite structure makes it easier to reach the electrodes as compared to more robust glass/metal systems [15,16]. Electric contacts are positioned on both sides of the fiber to avoid unwanted arc and are connected to a DC generator (Branderburg AlphaIII). At atmospheric pressure and in this planar configuration with the electrodes separated by a ~300 µm gap, a potential of 1–1.5 kV was enough to generate a discharge at the tip of the fiber assembly [17].

Pictures of the electric arc that can be observed in this configuration are given in Figure 2a. The discharge running between the two metallic electrodes passes in front of the extremity of the MMF in which the optical emission of the plasma is coupled. Light is guided toward the other end of the silica fiber terminated by an FC/PC connector, which is directly plugged to a compact fibered UV-Vis spectrometer (Avantes AvaSpec-ULS4096CL-EVO) operating in the 200–1100 nm spectral domain. Description of the experimental setup of this proof-of-principle measurement can be found in Figure 2a. A typical spectrum of a plasma generated in ambient air and registered using the developed device is depicted in Figure 2b. Only the main emission lines are referenced in this paper for clarity purposes. The signatures of species such as argon, nitrogen, oxygen, and water-related compounds can be identified according to literature [18], which proves the potential of this experimental setup for gas detection applications. On the spectrum of air, the second positive system of N_2_ (C^3^∏_u_ − B^3^∏_g_) is visible in the near-UV as well as N, O, and H_a_ atomic lines between 650 nm and 950 nm.

Then, to further grasp the potential of our optical detection device, we now proceed to overtake a more sophisticated task, namely identifying the main constituent of a probed atmosphere. Quantitative measurements based on plasma spectroscopy are usually extremely difficult and necessitate complex data treatment and analysis, which is not the subject in this study. For this reason, and given the precision of the spectrometer used, only qualitative characterizations were conducted. For the experiment, the most commonly available gases (nitrogen, oxygen, and argon) were used. A few Litters-per-minute flow is directed toward the probing extremity of the fiber assembly, which places the optical emission detection device in an N_2_, O_2_, and Ar-rich atmosphere. The recorded spectra are given, respectively, in Figure 3a–c. They can be easily differentiated depending on the probed chemical species. In each spectrum, the contribution of the pollutant, in our case the air, is detected. In nitrogen-rich discharge, the second positive system of N_2_, N, O, an H_a_ atomic lines are clearly identified. In argon-rich discharge, these lines were still present among the strong Ar atomic lines. In oxygen-rich plasma, the O atomic lines were much stronger than N and H_a_ atomic lines. When each of these specific atmospheres are probed, residual lines related to lingering species present in low concentration (hydrogenated compounds, air components) are still being picked up, which means the setup can be considered for trace detection. No signal is detected below 300 nm. This limitation is due to the increasing losses of the silica fiber toward shorter wavelengths. UV-transmitting optical patch cables are commercially available. However, they generally exhibit strong water absorptions in the NIR between 700 and 1600 nm. Additionally, despite being of great interest due to their technological maturity, silica fibers are not the best solution for UV transmission as the SiO_2_ glass matrix suffers from photo-induced defects formation when exposed to high frequency radiations. The use of ultra-violet (UV)-resistant fluorophosphate-based fibers with good transmission in this spectral domain [19] could be considered to help retrieve important information between 250 and 350 nm. To further improve the performances of the device, the numerical aperture of the collection fiber could also be optimized in order to collect more signal. Still, the device developed in this study shows good potential for real-time atmosphere monitoring.

## 3. Conclusions

We have demonstrated the potential of the combination of polymer/metal and silica fibers for the elaboration of an optical emission detector. We exploited the good fiber drawing ability of polymers to fabricate a hollow elongated structure with two embedded electrodes. This composite fiber was then assembled with a silica MMF to form the measurement system at the tip of which a plasma discharge was successfully generated using a 1–1.5 kV potential. Then, we demonstrated that the light collected from the glow discharge can be used to determine the main chemical constituent of a probed atmosphere through plasma spectroscopy. The assembly technique presented in this paper represents a cheap and simple route to functionalize existing commercial fibers and form compact multi-purpose hybrid devices since it is based on the highly scalable thermal drawing process. Several tens-of-meter of polymer-metal structure can be fabricated from a single preform, which yields hundreds of sections that can then be used in the type of assembly described in this work. In addition to the compatibility of the drawing process to industrial production, the materials involved in the functional cladding are inexpensive. Our results pave the way for integrating multiple electrodes in close vicinity with an optical waveguide allowing for relevant electro-optical stimulation or for applying homogeneous electric fields, which can be mandatory for optimal sensing. We believe many other configurations based on this low-priced combination of exotic composite elongated structures and common fibers can be considered for developing robust onboard or onsite sensing solutions suitable for operating in harsh environments as well as for endoscopic probing and more.

## Figures and Tables

**Figure 1 sensors-20-02353-f001:**
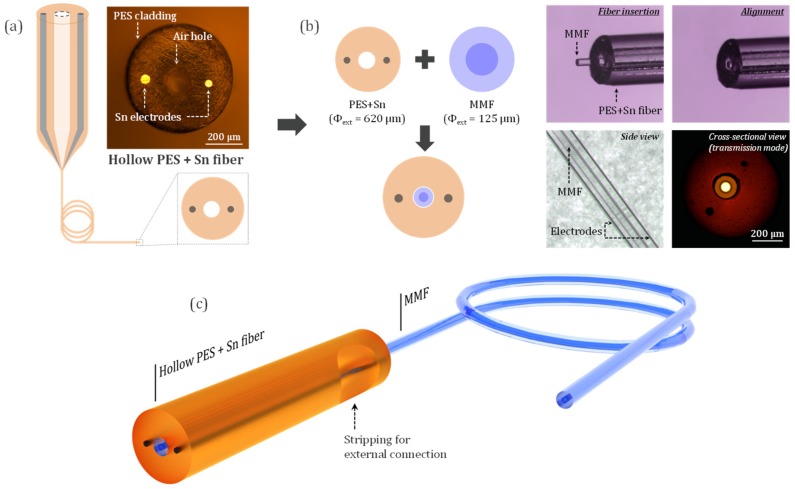
(**a**) Thermal drawing of hollow polymer-metal hybrid fibers. (**b**) Description of the assembly of a hollow polyether-sulfone (PES)+Sn fiber with a multimode-silica fiber. (**c**) 3D scheme of the optical emission detector device fabricated in this manner.

**Figure 2 sensors-20-02353-f002:**
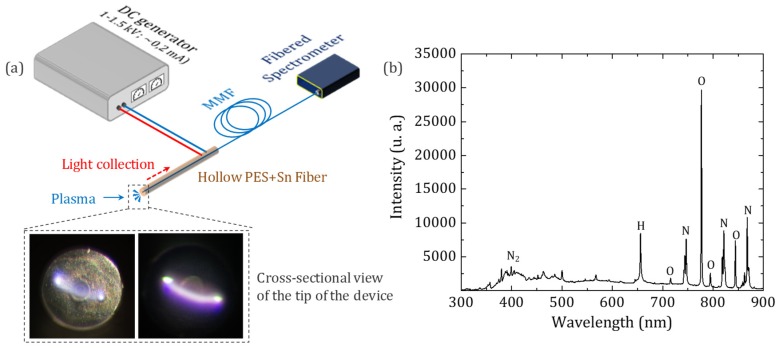
(**a**) Description of the experimental setup used for plasma spectroscopy involving an assembly of a polymer-metal and a silica fiber. The two pictures of the plasma (bottom left) generated at the tip of the assembly are taken in different lighting conditions: white light illumination (left) and in the dark (right). (**b**) A typical spectrum collected from this proof-of-principle device.

**Figure 3 sensors-20-02353-f003:**
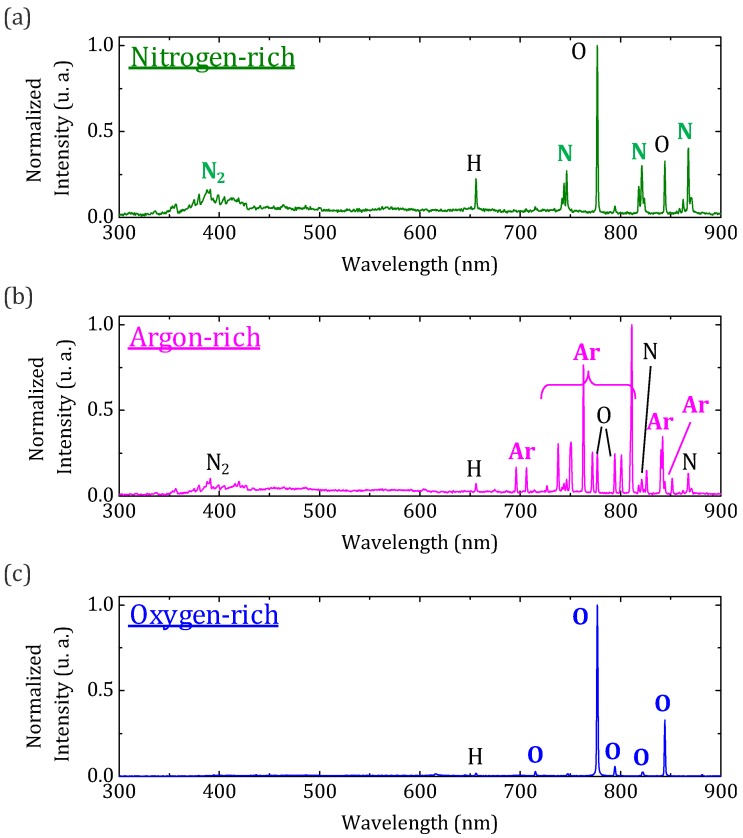
Typical emission spectra collected from the device placed under (**a**) nitrogen-rich, (**b**) argon-rich, and (**c**) oxygen-rich gas flow.

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
