# Peer review of "Optical Emission Detector Based on Plasma Discharge Generation at the Tip of a Multimaterial Fiber"

_sensors, 2020, doi:10.3390/s20082353_

Round 1

Reviewer 1 Report

The authors present an interesting idea in which an optical fibre is engineered as an optical emission detector with an integrated plasma discharge electrode system. However, I feel that this is a rather overly complex method of manufacturing the electrode system - they could have simply adhered thin (say 25um diameter) tungsten wires to the sides of the silica fibre for example to achieve a similar result in the first instance. 

I would therefore like to have seen a more expansive motivation for the use of a multimaterial approach to the fabrication of such hollow dual core composite structures. For example, if the authors had future plans for using a more advanced embodiment of this structure for hollow core optical waveguiding, then it would be a far more compelling reason for the preform manufacture. Otherwise it just seems like a very complicated way of creating simple electrodes. Perhaps an addition of a paragraph or two in the conclusions to motivate this work would make for a more complete manuscript.

Author Response

Response to Reviewer 1 comments

Point 1: The authors present an interesting idea in which an optical fibre is engineered as an optical emission detector with an integrated plasma discharge electrode system. However, I feel that this is a rather overly complex method of manufacturing the electrode system - they could have simply adhered thin (say 25um diameter) tungsten wires to the sides of the silica fibre for example to achieve a similar result in the first instance.

Response 1:

When consulting reported works in the literature concerning multimaterial fibers, it appears that adding electrodes to a glass fiber is not simple. The metallic parts are indeed either co-drawn within a glass preform (implying non-trivial material compatibility problematics) or added post fiber drawing (solid wire insertion or suction of liquid metal). In this context it seems that our method remains relevant as it meets the requirements of electro-optical fibered device microfabrication: besides the obvious insulation that must be provided over meters of fiber, it guaranties fine control of the position, size and even shape of the electrodes with high reproductivity.

Point 2: I would therefore like to have seen a more expansive motivation for the use of a multimaterial approach to the fabrication of such hollow dual core composite structures. For example, if the authors had future plans for using a more advanced embodiment of this structure for hollow core optical waveguiding, then it would be a far more compelling reason for the preform manufacture. Otherwise it just seems like a very complicated way of creating simple electrodes. Perhaps an addition of a paragraph or two in the conclusions to motivate this work would make for a more complete manuscript.

Response 2:

As pointed out by the reviewer, a more elegant way to implement the device described in this work would be to utilize a unique fiber featuring simultaneously waveguiding and electric functionalities. We are currently conducting investigations concerning this interesting topic. However, such development relies on direct preform-to-fiber drawing of polymer-glass-metal structures and still remains an important challenge implying non-trivial problematics. Among them are for instance the determination of thermally compatible materials (glass/metal/polymers) for fiber drawing, connection of the electrodes without compromising the waveguiding structures, optical losses of the involved glasses, and others. In this context, the device fabrication method presented here is in fact an intermediate step towards the direct preform-to-fiber drawing of hybrid structures, and seems to help overcome some of the difficulties mentioned above. More importantly it offers great freedom in the design of the polymer-metal cladding that is then combined with commercially available fibers with optimal optical transmission. For gas detection through plasma discharge generation, a simple cladding with only two metallic electrodes is required, which is not the case for other applications of electro-optic systems. For instance, the development of endoscopic probes usually involves the integration of 4 or more electrodes in the device combined with a waveguide for electro-optic stimulation and/or detection. For effective cancer treatment, the probes used in electrotherapy need the electrodes to be spatially arranged in specific geometries for the applied electric field to be homogeneous over the largest possible area.

We have added a sentence in the conclusion:

“Our results pave the way for the integration of multiple electrodes in close vicinity with an optical waveguide allowing for relevant electro-optical stimulation for instance or for the application of homogeneous electric fields, which can be mandatory for optimal sensing. “

Reviewer 2 Report

In this communication paper authors presented very interesting and to my best knowledge novel idea of integrated fiber optic tip with Sn electrodes and polymer cladding for plasma discharge based gas sensing. Plasma discharge based gas sensing is of course well known, however novelty of presented solution lies in the measurement tip which ensures the possibility of plasma generation via discharge directly next to the front of optical fiber what allows the miniaturization of the sensing device. The optic fibers with integrated electrodes are known from some other solutions like medical applications, however the idea of measuring the gas composition using such device seems to be novel for me (I did not found any papers which report similar solution). Advantage of this solution is also the possibility of changing the optical fibers for any commercial ones which my be chosen to adequate spectral range. 

Authors presented the technological details of tip fabrication and some qualitative examples of VIS-NIR spectra collected using the reported device which clearly presented their idea.

Paper is well written and organized and  clearly meets the Sensors journal scope. Keeping in mind that the paper is the communication which should briefly show the novel ideas in the field I am recommending this paper for publication.

The only inconsistency for me is the value of the voltage used during experiments - in line 122, page 3 and in the figure 2a authors talking about 1 kV, but in the conclusions (line 165, page 5) authors talking about 1.5 kV, please explain that.

Author Response

Response to Reviewer 2 comments

In this communication paper authors presented very interesting and to my best knowledge novel idea of integrated fiber optic tip with Sn electrodes and polymer cladding for plasma discharge based gas sensing. Plasma discharge based gas sensing is of course well known, however novelty of presented solution lies in the measurement tip which ensures the possibility of plasma generation via discharge directly next to the front of optical fiber what allows the miniaturization of the sensing device. The optic fibers with integrated electrodes are known from some other solutions like medical applications, however the idea of measuring the gas composition using such device seems to be novel for me (I did not found any papers which report similar solution). Advantage of this solution is also the possibility of changing the optical fibers for any commercial ones which my be chosen to adequate spectral range.

Authors presented the technological details of tip fabrication and some qualitative examples of VIS-NIR spectra collected using the reported device which clearly presented their idea.

Paper is well written and organized and clearly meets the Sensors journal scope. Keeping in mind that the paper is the communication which should briefly show the novel ideas in the field I am recommending this paper for publication.

Point 1: The only inconsistency for me is the value of the voltage used during experiments - in line 122, page 3 and in the figure 2a authors talking about 1 kV, but in the conclusions (line 165, page 5) authors talking about 1.5 kV, please explain that.

Response 1: We thank the reviewer for having this inconsistency pointed out. The reason for it is that the applied potential could not be precisely measured with the setup that was used here. It is estimated to be 1-1.5 kV. Future work will consist in precisely determining the involved currents and potentials in order to fully understand the discharge generation dynamics observed here. Before this, we prefer to remain precautious and only indicate approximate values.

Page 3, line 112: “…a potential superior to 1 kV was enough to generate a discharge at the tip of the fiber assembly “

We corrected the sentence: “At atmospheric pressure and in this planar configuration with the electrodes separated by a ~300 µm gap, a potential of 1-1.5 kV was enough to generate a discharge at the tip of the fiber”

Page 4, Figure 2a:

We corrected the given values of current and potential:

“DC generator (~1 kV ; 0,2 mA)” was replaced by: “DC generator (1-1.5 kV ; ~0,2 mA)”

Page 5, line 165: “… was successfully generated using a 1.5 kV potential.”

We corrected the sentence: “This composite fiber was then assembled with a silica MMF to form the measurement system at the tip of which a plasma discharge was successfully generated using a 1-1.5 kV potential.”